# Overnutrition and Lipotoxicity: Impaired Efferocytosis and Chronic Inflammation as Precursors to Multifaceted Disease Pathogenesis

**DOI:** 10.3390/biology13040241

**Published:** 2024-04-06

**Authors:** Vivek Mann, Alamelu Sundaresan, Shishir Shishodia

**Affiliations:** Department of Biology, Texas Southern University, Houston, TX 77004, USA; vivek.mann@tsu.edu (V.M.); alamelu.sundaresan@tsu.edu (A.S.)

**Keywords:** overnutrition, lipotoxicity, oxidative stress, efferocytosis, chronic inflammation

## Abstract

**Simple Summary:**

Overnutrition, characterized by an excessive caloric intake, often leads to lipotoxicity, a condition where lipids accumulate abnormally in non-adipose tissues. This phenomenon contributes to impaired efferocytosis, the process by which cells remove dead or dying cells, and subsequently triggers chronic inflammation. These interconnected factors serve as precursors to a multitude of disease pathologies. The overabundance of nutrients overwhelms cellular mechanisms, disrupting the delicate balance required for proper efferocytosis. As a result, apoptotic cells linger, perpetuating inflammation and triggering a cascade of detrimental effects on tissue function and homeostasis. Chronic inflammation, a hallmark of various diseases, including cardiovascular disorders, diabetes, and neurodegenerative conditions, underscores the significance of understanding the underlying mechanisms linking overnutrition, impaired efferocytosis, and disease pathogenesis. By elucidating these pathways, researchers aim to develop targeted interventions to mitigate the adverse health outcomes associated with overnutrition-induced lipotoxicity, offering potential avenues for prevention and treatment to combat multifaceted diseases.

**Abstract:**

Overnutrition, driven by the consumption of high-fat, high-sugar diets, has reached epidemic proportions and poses a significant global health challenge. Prolonged overnutrition leads to the deposition of excessive lipids in adipose and non-adipose tissues, a condition known as lipotoxicity. The intricate interplay between overnutrition-induced lipotoxicity and the immune system plays a pivotal role in the pathogenesis of various diseases. This review aims to elucidate the consequences of impaired efferocytosis, caused by lipotoxicity-poisoned macrophages, leading to chronic inflammation and the subsequent development of severe infectious diseases, autoimmunity, and cancer, as well as chronic pulmonary and cardiovascular diseases. Chronic overnutrition promotes adipose tissue expansion which induces cellular stress and inflammatory responses, contributing to insulin resistance, dyslipidemia, and metabolic syndrome. Moreover, sustained exposure to lipotoxicity impairs the efferocytic capacity of macrophages, compromising their ability to efficiently engulf and remove dead cells. The unresolved chronic inflammation perpetuates a pro-inflammatory microenvironment, exacerbating tissue damage and promoting the development of various diseases. The interaction between overnutrition, lipotoxicity, and impaired efferocytosis highlights a critical pathway through which chronic inflammation emerges, facilitating the development of severe infectious diseases, autoimmunity, cancer, and chronic pulmonary and cardiovascular diseases. Understanding these intricate connections sheds light on potential therapeutic avenues to mitigate the detrimental effects of overnutrition and lipotoxicity on immune function and tissue homeostasis, thereby paving the way for novel interventions aimed at reducing the burden of these multifaceted diseases on global health.

## 1. Introduction

The pathogenesis of various diseases involves complex interactions among different factors, including injury stimuli, immune responses, and metabolic imbalances caused by overnutrition and lipotoxicity [1]. Diseases can often originate from tissue damage or injury. This damage can be caused by physical trauma, infections, toxins, or other harmful stimuli. In response to injury, cells release signaling molecules, such as cytokines and chemokines, which play a crucial role in immune responses and tissue repair [2]. The immune system is a vital player in the body’s defense against infections and other harmful agents [3]. However, an overactive or misdirected immune response can contribute to the development of various diseases. Autoimmune disorders, for example, result from the immune system mistakenly attacking the body’s own tissues. Metabolism refers to the set of chemical processes that occur within living organisms to maintain life [4]. Disturbances to metabolic pathways can contribute to disease. Overnutrition, or an excessive intake of nutrients, can lead to metabolic disorders such as obesity, diabetes, and cardiovascular diseases [5].

Lipotoxicity refers to the harmful effects of excessive lipid accumulation in non-adipose tissues, which can contribute to insulin resistance and other metabolic problems. The interplay between injury stimuli, immune responses, and metabolic imbalances can create a cycle of events that perpetuates and exacerbates disease [6]. For example, chronic inflammation resulting from an immune response to persistent injury can contribute to metabolic dysfunction and tissue damage [2]. Comprehending the complex relationships among these factors is essential for crafting efficacious treatments and preventive measures for a multitude of diseases. Moreover, genetic predisposition, environmental factors, and lifestyle choices also play roles in disease pathogenesis, adding further layers of complexity to the overall picture [7]. Researchers and healthcare professionals continually work to unravel these complexities to improve diagnostics, treatment strategies, and overall disease management. This review aims to provide a summary of the current understanding of how these elements interplay and contribute to the development and progression of diseases.

## 2. Overnutrition and Lipotoxicity

Overnutrition and lipotoxicity are closely related concepts that play significant roles in the pathogenesis of various diseases, particularly those associated with metabolic dysfunction [6]. Chronic overnutrition promotes adipose tissue expansion and ectopic lipid accumulation in non-adipose tissues, leading to an overflow of free fatty acids and lipid metabolites. This lipotoxic environment induces cellular stress and inflammatory responses, contributing to insulin resistance, dyslipidemia, and metabolic syndrome [8]. Overnutrition occurs when the intake of nutrients, especially calories, exceeds the body’s metabolic needs. This imbalance can lead to the accumulation of excess energy in the form of fat [9]. The modern lifestyle, characterized by easily accessible high-calorie foods and sedentary behavior, has contributed to a rise in overnutrition-related issues, such as obesity, type 2 diabetes, and cardiovascular diseases [10].

Lipotoxicity pertains to the adverse consequences arising from the excessive buildup of lipids in non-adipose tissues, including but not limited to the liver, pancreas, heart, and muscles [11]. When cells are exposed to elevated levels of free fatty acids, it can lead to cellular dysfunction, inflammation, and apoptosis. Lipotoxicity is particularly relevant in the context of insulin resistance, a condition where cells become less responsive to the effects of insulin, a hormone that regulates blood sugar, which is a common feature in obesity and type 2 diabetes [12]. The connection between overnutrition and lipotoxicity is evident in the following ways:

### 2.1. Excess Caloric Intake

Overnutrition, which involves consuming an excessive number of calories and nutrients, often leads to an increased intake of dietary fats. When the body receives more calories than it needs for energy expenditure, the excess dietary fats can be stored in adipose tissue. This accumulation of fat can result in obesity and contribute to various metabolic disorders [13]. One consequence of overnutrition, especially an excessive intake of saturated fats, is an elevation in circulating free fatty acids in the bloodstream [14]. Free fatty acids are molecules released when triglycerides are broken down, and they serve as a source of energy for various tissues [15]. However, an excess of circulating free fatty acids, particularly saturated fatty acids, can lead to lipotoxicity [16]. Obesity is a complex condition with various phenotypes, which can be classified based on different criteria including etiology, distribution of fat, metabolic profile, and associated comorbidities [17]. The consumption of ultra-processed foods is a significant factor contributing to overnutrition and obesity in many populations worldwide [18]. Ultra-processed foods are typically high in refined carbohydrates, added sugars, unhealthy fats, and salt, while often lacking in essential nutrients like fiber, vitamins, and minerals [18,19]. Ultra-processed foods tend to be energy-dense, meaning they provide a high number of calories relative to their weight. This can lead to excessive calorie intake, especially when portion sizes are not controlled. Regular consumption of calorie-dense foods contributes to a positive energy balance, leading to weight gain over time [20].

Ultra-processed foods often lack dietary fiber and protein, which are important for promoting feelings of fullness and satiety. As a result, individuals may consume larger quantities of these foods before feeling satisfied, leading to overeating and excessive calorie intake [21]. Many ultra-processed foods contain high levels of added sugars, including sugary beverages, snacks, and desserts [18]. Excessive sugar consumption can contribute to weight gain and obesity by providing empty calories and promoting insulin resistance, leading to metabolic dysregulation and increased fat storage [22].

Ultra-processed foods often contain unhealthy fats, such as trans fats and saturated fats, which can negatively impact cardiovascular health and contribute to weight gain. These fats are commonly found in fried foods, processed meats, and baked goods, all of which are prevalent in the ultra-processed food supply [23]. Ultra-processed foods are often convenient and readily available, making them a convenient option for busy individuals or those with limited access to fresh, whole foods [21]. However, frequent consumption of these foods can lead to a reliance on highly processed and nutritionally poor options, contributing to poor dietary habits and an increased risk of obesity [24].

Regular consumption of ultra-processed foods can disrupt normal eating patterns and promote unhealthy dietary behaviors, such as mindless snacking, emotional eating, and a reliance on processed convenience foods [20]. These behaviors can further contribute to overnutrition and obesity over time. Addressing the overconsumption of ultra-processed foods requires comprehensive strategies aimed at promoting healthier dietary habits, improving food environments, and increasing access to nutritious whole foods [18]. Public health interventions, policy changes, nutrition education programs, and industry regulations are all essential components of efforts to reduce the prevalence of overnutrition and obesity associated with the consumption of ultra-processed foods [25]. Patients can be put into the following categories according to their metabolic profile (Figure 1):

MUNO: Metabolically Unhealthy Non-Obese. /MONW: Metabolically Obese Normal Weight. These terms are used to describe individuals who have a normal body weight according to traditional measures such as BMI (Body Mass Index) but exhibit metabolic abnormalities like those found in obese individuals [26]. This means they may have issues like insulin resistance, high blood pressure, high cholesterol levels, and other markers of metabolic syndrome despite not being classified as obese based on their BMI alone [27]. The distinguishing factor between MUNO and MONW lies in the presence or absence of metabolic abnormalities. MUNO individuals have these abnormalities, while MONW individuals do not. Both groups have a normal body weight according to their BMI, but their metabolic profiles differ [26].

MHNO: Metabolically Healthy Non-Obese. This term refers to individuals who have a normal body weight according to their BMI and exhibit a favorable metabolic profile. They do not have the metabolic abnormalities typically associated with obesity, such as insulin resistance, high blood pressure, or dyslipidemia [28]. MHNO individuals have a favorable metabolic profile, unlike those classified as MUNO or MONW. Like MUNO and MONW, MHNO individuals have a normal body weight according to their BMI [29].

All of these terms deal with the relationship between weight status and metabolic health, but they differ in the presence or absence of metabolic abnormalities despite normal weight. MUNO and MONW describe individuals with metabolic abnormalities and normal weight, whereas MHNO refers to those with both normal weight and a favorable metabolic profile [30]. Maintaining a balanced and healthy diet is crucial for preventing these adverse effects.

### 2.2. Adipose Tissue Dysfunction

Overnutrition can lead to dysfunctional adipose tissue, which plays a role in regulating lipid metabolism. As adipose tissue becomes overwhelmed, it may release excessive amounts of fatty acids into the bloodstream, contributing to lipotoxic effects in other tissues [31,32]. Adipose tissue, commonly known as fat tissue, is not just a passive storage depot for excess energy; it is an active endocrine organ involved in regulating energy balance and metabolism [33]. In a state of overnutrition, where there is an excess intake of calories, adipose tissue can become dysfunctional. This dysfunction can lead to an imbalance in the release of adipokines (hormones produced by adipose tissue) and an increase in the release of free fatty acids into the bloodstream [34]. As adipose tissue becomes overwhelmed, it may release excessive amounts of fatty acids, contributing to elevated circulating levels of these molecules. The increased release of fatty acids from adipose tissue can have several consequences, including the following:Lipotoxicity in Other Tissues: Elevated levels of circulating free fatty acids can contribute to lipotoxic effects in various tissues like the liver, pancreas, and skeletal muscle (Figure 2). This lipotoxicity can impair cellular function and contribute to insulin resistance, inflammation, and other metabolic disturbances [35,36].Insulin Resistance: Excess fatty acids can disrupt insulin signaling, leading to insulin resistance. This condition impairs glucose uptake and metabolism, resulting in elevated blood sugar levels (hyperglycemia). Insulin resistance is a key factor in the development of metabolic disorders like type 2 diabetes, influenced by both overnutrition and lipotoxicity [37].Inflammation: Lipotoxicity is associated with an inflammatory response. Cytokines play a crucial role in inflammation, which is the body’s response to injury, infection, or other stimuli. They are small proteins secreted by various cells, including immune cells, and act as signaling molecules to regulate immune responses, inflammation, and other physiological processes [38]. In the context of inflammation, cytokines can be pro-inflammatory or anti-inflammatory, and their balance is essential for maintaining immune homeostasis. Pro-inflammatory cytokines promote inflammation by inducing vasodilation, increasing vascular permeability, recruiting immune cells to the site of injury or infection, and activating immune responses [39]. Examples include interleukin-1 (IL-1), interleukin-6 (IL-6), tumor necrosis factor-alpha (TNF-α), and interferon-gamma (IFN-γ) [40]. Pro-inflammatory cytokines play a critical role in the initial response to pathogens and tissue damage [39]. In contrast, anti-inflammatory cytokines help to resolve inflammation and maintain an immune balance by inhibiting pro-inflammatory responses and promoting tissue repair and regeneration. Examples include interleukin-10 (IL-10) and transforming growth factor-beta (TGF-β). Anti-inflammatory cytokines are essential for preventing excessive inflammation and tissue damage [41].

Lipid peroxidation (LPO) is a process in which free radicals oxidize lipids, leading to the production of reactive lipid species that can damage cell membranes and other cellular components [42]. Ferroptosis is a form of regulated cell death characterized by the iron-dependent accumulation of lipid peroxides and is distinct from other forms of cell death such as apoptosis or necrosis [43]. The importance of LPO-induced ferroptotic cell death in disease progression has been increasingly recognized, particularly in diseases characterized by oxidative stress, inflammation, and tissue damage [44]. Some of the diseases where LPO-induced ferroptosis plays a significant role include the following:

Ferroptosis, a form of cell death characterized by lipid peroxidation (LPO), is increasingly recognized as a contributing factor to neuronal cell death in neurodegenerative diseases like Alzheimer’s, Parkinson’s, and Huntington’s diseases [45]. The common features of oxidative stress and inflammation in these disorders suggest that LPO-induced ferroptosis may worsen neuronal damage and disease progression [43,46]. In cancer treatment, ferroptosis has emerged as a potential therapeutic strategy, particularly in resistant cancers, as tumor cells often display heightened sensitivity to ferroptosis due to their metabolic demands and reliance on iron-dependent processes [46]. Targeting LPO-induced ferroptosis in cancer cells offers a promising avenue for selective cancer therapy [46]. Additionally, during an ischemia-reperfusion injury, which is often seen in conditions like myocardial infarction or stroke, restoring blood flow to ischemic tissues can trigger oxidative stress and inflammation, leading to tissue damage and cell death. LPO-induced ferroptosis has been implicated in exacerbating tissue injury during ischemia-reperfusion, suggesting that targeting this process may hold therapeutic potential for mitigating tissue damage in such conditions [47].

Autophagy and mitophagy are vital cellular processes responsible for degrading and recycling damaged organelles, especially mitochondria, to maintain cellular balance and prevent damage [48]. Autophagy removes damaged components, while mitophagy specifically targets defective mitochondria. Excessive lipid accumulation can hinder autophagy, disrupting the removal of damaged organelles and proteins [49]. Lipid overload also interferes with the fusion of autophagosomes with lysosomes, hampering their ability to degrade and recycle materials, contributing to cellular dysfunction and lipotoxicity [50]. Lipotoxicity induces mitochondrial dysfunction, marked by impaired respiration, increased reactive oxygen species (ROS) production, and mitochondrial membrane depolarization [51]. Dysfunctional mitochondria are normally cleared via mitophagy, but in cases of overnutrition and lipotoxicity, this process may be compromised, leading to the buildup of damaged mitochondria and worsening cellular dysfunction and oxidative stress [52]. Autophagy plays a crucial role in lipid metabolism by facilitating lipid droplet degradation (lipophagy) and regulating lipid synthesis and storage. Impaired autophagy in overnutrition and lipotoxic conditions disrupts lipid metabolism, leading to lipid droplet accumulation and exacerbating cellular lipid overload [53]. Overnutrition and lipotoxicity can impair both autophagy and mitophagy, resulting in the buildup of damaged organelles, disrupted lipid metabolism, and cellular dysfunction [54].

Metabolic Syndrome: Overnutrition contributes to a cluster of conditions known as metabolic syndrome, including abdominal obesity, insulin resistance, high blood pressure, and dyslipidemia. Lipotoxicity exacerbates these conditions, leading to obesity, type 2 diabetes, and non-alcoholic fatty liver disease (NAFLD). Chronic inflammation further worsens metabolic dysfunction [55].

Lipotoxicity, resulting from excessive lipid accumulation outside adipose tissues, extends beyond metabolic disorders like obesity and diabetes, impacting conditions such as depression, sleep apnea, and infertility [56]. In depression, lipotoxicity contributes to chronic inflammation and oxidative stress, disrupting neuronal function and neurotransmitter balance, potentially exacerbating depressive symptoms [57]. Sleep apnea, which is strongly linked to obesity, involves lipotoxicity-induced inflammation and oxidative stress, contributing to structural changes in airway tissues, and leading to an increased susceptibility to airway collapse during sleep [58]. In infertility, lipotoxicity disrupts endocrine function, altering sex hormone levels and insulin resistance, affecting ovarian and testicular function, impairing fertility [59]. Lipid accumulation damages reproductive tissues, impacting gamete quality and function, hindering folliculogenesis, spermatogenesis, and embryo implantation [60]. 

In obesity, adipose tissue expands to store excess energy, resulting in changes in the adipose tissue microenvironment. Additionally, overnutrition and lipotoxicity can cause fat accumulation in the liver, leading to Non-Alcoholic Fatty Liver Disease (NAFLD) [61]. NAFLD encompasses various stages of liver conditions, ranging from simple fatty liver to more severe forms like non-alcoholic steatohepatitis (NASH) with inflammation and fibrosis [62]. Advanced NAFLD, especially in individuals with cirrhosis, increases the risk of hepatocellular carcinoma (HCC), a form of liver cancer [63] (Figure 3). Risk factors for NAFLD include obesity, insulin resistance, type 2 diabetes, metabolic syndrome, and lifestyle factors such as a sedentary lifestyle and poor dietary habits [64]. Early detection and management of risk factors, including lifestyle modifications such as a healthy diet, regular exercise, and weight management, are crucial for preventing NAFLD progression and reducing the risk of complications [65]. 

### 2.3. Impaired Efferocytosis/Oxidative Stress/Immune Response

The molecular mechanisms underlying impaired efferocytosis and chronic inflammation involve a complex interplay between various cellular processes and signaling pathways [66]. Efferocytosis begins with the recognition and binding of apoptotic cells by phagocytes, primarily macrophages [67]. This process involves the interaction between “eat-me” signals on apoptotic cells and receptors on phagocytes. Examples of eat-me signals include phosphatidylserine (PS) exposure on the outer leaflet of the plasma membrane of apoptotic cells [68,69]. Receptors such as TAM receptors (Tyro3, Axl, and MerTK) and scavenger receptors on macrophages bind to these signals to facilitate engulfment [70]. Following recognition and binding, phagocytes engulf apoptotic cells through actin cytoskeleton rearrangements and membrane remodeling. This step requires the activation of various signaling pathways, including Rho GTPases, phosphoinositide 3-kinase (PI3K), and Rac1, which regulate the cytoskeletal dynamics and membrane protrusions necessary for phagocytic cup formation and closure [71]. Once internalized, engulfed apoptotic cells undergo degradation within phagocytes through lysosomal fusion and enzymatic digestion [72]. This process is regulated by intracellular signaling cascades, including the activation of protein kinase C (PKC), mitogen-activated protein kinases (MAPKs), and phospholipase A2 (PLA2), which coordinate phagosome maturation and the degradation of apoptotic cell components [73,74].

Efferocytosis is typically associated with the induction of anti-inflammatory signaling pathways to dampen inflammation and promote tissue repair. This includes the production of anti-inflammatory cytokines such as transforming growth factor-beta (TGF-β) and interleukin-10 (IL-10) by phagocytes, which inhibit pro-inflammatory cytokine production and promote the resolution of inflammation [75]. Impaired efferocytosis can arise from defects in phagocyte function, including alterations to receptor expression or the signaling pathways which are critical for apoptotic cell clearance. For example, deficiencies in TAM receptor signaling or defects in cytoskeletal regulators can impair phagocyte-mediated engulfment of apoptotic cells [70].

Chronic inflammation can disrupt efferocytosis by creating an inflammatory microenvironment that impairs phagocyte function and promotes pro-inflammatory responses [75]. Inflammatory mediators such as tumor necrosis factor-alpha (TNF-α) and interferon-gamma (IFN-γ) can inhibit efferocytosis by downregulating phagocytic receptor expression or promoting the pro-inflammatory activation of phagocytes [76]. Impaired efferocytosis and chronic inflammation are intertwined processes that contribute to the pathogenesis of various diseases [75]. 

In impaired efferocytosis and chronic inflammation, various immune cells, particularly macrophages, play pivotal roles. The dysregulation of various immune cells, particularly macrophages, can contribute to impaired efferocytosis and chronic inflammation [77]. Macrophages play a crucial role in efferocytosis, the clearance of dying or dead cells from tissues, but the dysregulation that occurs within chronic inflammation can impair this process, leading to the accumulation of apoptotic cells and perpetuating inflammation [75]. Various types of macrophages are involved:

Resident tissue macrophages are derived from embryonic precursors, which maintain tissue homeostasis and immune surveillance. Impaired efferocytosis can lead to chronic inflammation and tissue damage [78]. Inflammatory macrophages are recruited to sites of inflammation from circulating monocytes. These macrophages may exhibit impaired efferocytosis under inflammatory conditions, perpetuating inflammation [79]. Foam cells are found in atherosclerotic plaques, where they can accumulate lipid droplets and exhibit impaired efferocytosis due to lipid overload, contributing to plaque instability and inflammation [80]. Tumor-associated macrophages (TAMs) are present within tumor microenvironments, and they demonstrate impaired efferocytosis, promoting tumor progression and immune evasion [81]. Peritoneal macrophages inhabit the peritoneal cavity, and compromised efferocytosis due to these macrophages is a factor in conditions such as peritonitis and inflammatory disorders affecting the abdomen [82]. Alveolar macrophages, situated within lung alveoli, play a crucial role in maintaining respiratory health. Dysfunction in their efferocytosis process can exacerbate pulmonary diseases like COPD and asthma [83].

Additionally, neutrophils, while not primarily involved in efferocytosis, release pro-inflammatory mediators and reactive oxygen species, contributing to tissue damage and inflammation if dysregulated [84]. Dendritic cells, T cells, B cells, mast cells, and natural killer (NK) cells also play roles in chronic inflammation by promoting immune responses, sustaining inflammation, and contributing to tissue damage when dysregulated [2]. Dysfunctional immune cell responses can perpetuate chronic inflammation and contribute to tissue damage in various diseases [2,3].

Efferocytosis and mitophagy are two interconnected cellular processes involved in the regulation of immune responses, particularly in the context of macrophage and immune cell function during disease pathogenesis [85]. Efferocytosis and mitophagy play complementary roles in dampening inflammation and resolving immune responses [75]. Efferocytosis prevents the accumulation of apoptotic cells and the release of pro-inflammatory cellular contents, whereas mitophagy limits the release of DAMPs from dysfunctional mitochondria [75,86]. Together, efferocytosis and mitophagy contribute to the resolution of inflammation and the promotion of tissue repair following injury or infection [75].

Emerging evidence suggests that there is crosstalk between the efferocytosis and mitophagy pathways through their shared molecular components and regulatory mechanisms [87]. For example, some molecules involved in the recognition and engulfment of apoptotic cells by macrophages, such as phosphatidylserine receptors and scavenger receptors, also play roles in the regulation of autophagy and mitophagy [72]. Furthermore, defects in efferocytosis or mitophagy can disrupt immune homeostasis and exacerbate inflammation, contributing to the pathogenesis of various diseases [75]. Dysregulated efferocytosis and mitophagy are implicated in the pathogenesis of numerous diseases, including autoimmune disorders, atherosclerosis, neurodegenerative diseases, and cancer [88]. In conditions where efferocytosis or mitophagy is impaired, such as in aging or chronic inflammatory diseases, the accumulation of apoptotic cells or dysfunctional mitochondria can promote inflammation and tissue damage [89].

Oxidative stress refers to an imbalance between the production of reactive oxygen species (ROS) and the body’s ability to neutralize or repair the damage caused by these harmful molecules [90]. Reactive oxygen species (ROS) are highly reactive molecules containing oxygen, such as superoxide anion (O_2_•−), hydrogen peroxide (H_2_O_2_), and hydroxyl radicals (•OH). ROS are produced as natural byproducts of cellular metabolism, primarily in mitochondria, peroxisomes, and cytoplasmic enzymes like NADPH oxidases [91]. While ROS serve important roles in cell signaling, they can also lead to oxidative stress when produced in excess or when the cellular antioxidant defense mechanisms are overwhelmed [92]. In the domain of lipotoxicity, excessive lipid buildup in tissues beyond the adipose tissues can initiate mitochondrial dysfunction, inflammation, and oxidative stress, leading to an escalation in reactive oxygen species (ROS) generation [93].

Lipotoxicity often leads to mitochondrial dysfunction, which is characterized by impaired electron transport chain (ETC) activity, reduced ATP production, and increased electron leakage, resulting in elevated ROS generation within the mitochondria [93,94]. Excessive ROS production overwhelms the cellular antioxidant defense systems, including enzymes like superoxide dismutase (SOD), catalase, and glutathione peroxidase, as well as non-enzymatic antioxidants such as glutathione and vitamins C and E [95]. This imbalance between ROS production and antioxidant capacity leads to oxidative stress, causing damage to lipids, proteins, and DNA within the cell [96].

ROS can initiate lipid peroxidation, a chain reaction that damages cellular membranes by oxidizing polyunsaturated fatty acids (PUFAs) [97]. Lipid peroxidation generates lipid hydroperoxides and reactive aldehydes, such as malondialdehyde (MDA) and 4-hydroxynonenal (4-HNE), which can further exacerbate cellular damage and inflammation [42]. ROS serve as signaling molecules that activate inflammatory pathways, such as nuclear factor-kappa B (NF-κB) and mitogen-activated protein kinases (MAPKs), leading to the production of pro-inflammatory cytokines and chemokines [98]. Chronic inflammation further promotes lipotoxicity by perpetuating tissue damage and dysfunction [99].

Lipotoxicity-induced ROS production can also cause ER stress, leading to unfolded protein response (UPR) activation [100]. ER stress and UPR contribute to cellular dysfunction and apoptosis, further exacerbating tissue damage and dysfunction in lipotoxic conditions [100]. Prolonged exposure to elevated ROS levels can lead to cellular dysfunction and apoptosis, particularly in cells sensitive to oxidative stress, such as hepatocytes, cardiomyocytes, and pancreatic β-cells [101]. ROS play a central role in the pathogenesis of lipotoxicity by promoting mitochondrial dysfunction, oxidative stress, lipid peroxidation, inflammation, ER stress, cellular dysfunction, and apoptosis [102]. It is important to recognize the intricate connections between impaired efferocytosis, oxidative stress, and immune response to unravel the mechanisms underlying various diseases and to develop targeted therapeutic interventions. 

## 3. Linking Impaired Efferocytosis to Chronic Inflammation 

Chronic inflammation resulting from impaired efferocytosis is a condition where the normal process of clearing apoptotic cells is compromised, leading to the persistence of dead or dying cells in tissues [75]. This failure to efficiently remove apoptotic cells can contribute to a sustained and unresolved inflammatory response. The unresolved chronic inflammation perpetuates a pro-inflammatory microenvironment, exacerbating tissue damage and promoting the development of various diseases [103]. This is how a deficiency in efferocytosis contributes to the persistence of chronic inflammation:

### 3.1. Accumulation of Apoptotic Cells

In a healthy immune response, phagocytes like macrophages swiftly recognize and clear apoptotic cells, preventing the release of pro-inflammatory cellular contents. Impaired efferocytosis can lead to the accumulation of apoptotic cells in tissues, impacting tissue homeostasis and various physiological and pathological conditions [103]. When apoptotic cells are not efficiently cleared, their accumulation can trigger inflammation, potentially leading to autoimmune reactions and tissue damage, as observed in autoimmune disorders [104]. The buildup of apoptotic cells has been implicated in various diseases, including neurodegenerative diseases and certain cancers [105]. However, apoptotic cells also release anti-inflammatory signals that aid in resolving inflammation and promoting tissue repair [106]. The balance between apoptosis and the effective removal of apoptotic cells by phagocytes, particularly macrophages, is crucial for preserving tissue balance and preventing pathological conditions associated with apoptotic cell accumulation [107].

### 3.2. Secondary Necrosis

Secondary necrosis, also known as post-apoptotic necrosis, refers to the process by which apoptotic cells undergo further degradation and disintegration following incomplete clearance by phagocytes [108]. When apoptotic cells are not efficiently engulfed and removed by phagocytic, they progress to secondary necrosis, leading to the release of their cellular contents into the extracellular environment. This can trigger inflammation and contribute to the pathogenesis of various diseases, particularly those associated with impaired efferocytosis [109]. Examples of diseases where secondary necrosis contributes to disease progression due to impaired efferocytosis include the following: In autoimmune diseases like systemic lupus erythematosus (SLE) and rheumatoid arthritis (RA), impaired efferocytosis results in the buildup of apoptotic cells, leading to secondary necrosis. This release of autoantigens activates autoreactive immune cells, worsening autoimmune responses and tissue damage [110]. In atherosclerotic plaques, deficient efferocytosis causes apoptotic cell accumulation, promoting secondary necrosis and releasing pro-inflammatory molecules, exacerbating plaque instability and atherosclerosis progression [111]. In chronic obstructive pulmonary disease (COPD), impaired efferocytosis leads to apoptotic cell buildup in the lungs, exacerbating inflammation, mucus production, and tissue remodeling. Secondary necrosis further worsens lung damage and inflammation [112]. In non-alcoholic fatty liver disease (NAFLD) and non-alcoholic steatohepatitis (NASH), impaired efferocytosis causes apoptotic hepatocyte accumulation, leading to secondary necrosis and the release of pro-inflammatory molecules, contributing to hepatic inflammation, fibrosis, and disease progression [113]. In neurodegenerative diseases such as Alzheimer’s and Parkinson’s, impaired efferocytosis results in apoptotic cell accumulation in the brain, triggering neuroinflammation and the release of neurotoxic molecules, worsening neuronal damage and disease advancement [114]. Impaired efferocytosis can lead to the accumulation of apoptotic cells and secondary necrosis, contributing to inflammation, tissue damage, and disease progression in various pathological conditions, including autoimmune diseases, atherosclerosis, COPD, NASH, and neurodegenerative diseases [88]. Understanding the mechanisms that underline secondary necrosis is crucial for investigating potential therapeutic interventions in conditions where this process may be disrupted.

### 3.3. Activation of Inflammatory Pathways

The presence of uncleared apoptotic cells and cellular debris can activate inflammatory pathways, leading to sustained inflammation even without initial infectious or injurious stimuli [115]. Inflammation, a tightly regulated response to various stimuli like infections or tissue damage, involves the recognition of pathogen-associated molecular patterns (PAMPs) or damage-associated molecular patterns (DAMPs) [2]. The activation of inflammasomes, protein complexes sensing cellular stress, can initiate inflammation by promoting the release of pro-inflammatory cytokines like interleukin-1 beta (IL-1β) [116]. Pro-inflammatory cytokines recruit immune cells to the site of infection or injury, inducing migration via chemotaxis, facilitating phagocytic activity that is crucial for resolving infections and promoting tissue repair [38]. Mechanisms exist to resolve inflammation, involving anti-inflammatory cytokines and specialized pro-resolving lipid mediators; however, dysregulation can lead to chronic inflammation, which is implicated in autoimmune disorders, chronic inflammatory diseases, and certain cancers [117]. Understanding the delicate balance of inflammatory pathways is crucial for maintaining immune homeostasis and preventing inappropriate or chronic inflammation, which contributes to various diseases [118].

## 4. Chronic Inflammation and Its Association with Diseases

Chronic inflammation, a prolonged and dysregulated immune response, can detrimentally affect the body over time, contrasting with acute inflammation, a normal protective response to injury or infection [119]. Conditions such as atherosclerosis, rheumatoid arthritis, neurodegenerative disorders, and cancer have been linked to chronic inflammation due to impaired efferocytosis [120]. For instance, in atherosclerosis, impaired efferocytosis can lead to apoptotic cell accumulation in plaques, promoting inflammation and plaque progression, increasing the risk of cardiovascular events [121]. In rheumatoid Arthritis, inefficient apoptotic cell clearance in joints triggers an immune response, contributing to chronic inflammation, joint damage, and pain [122]. In neurodegenerative disorders, impaired efferocytosis in the central nervous system leads to apoptotic neuron accumulation, activating microglia and contributing to chronic inflammation, neuronal damage, and disease progression [75].

In cancer, chronic inflammation creates a microenvironment conducive to oncogenic mutations, promoting tumor growth through mechanisms such as genomic instability and DNA damage induced by reactive oxygen and nitrogen species [123,124]. Tumor immune escape refers to the ability of cancer cells to evade detection and elimination by the immune system, allowing them to proliferate and establish tumors. This process involves multiple mechanisms that suppress immune responses and promote tumor growth [125]. Chronic inflammation and impaired efferocytosis can contribute to tumor immune escape by creating an immunosuppressive microenvironment conducive to tumor progression [126]. Chronic inflammation is a hallmark of cancer development and progression. Inflammatory mediators produced in response to tissue damage, infection, or other insults can promote tumor initiation, angiogenesis, invasion, and metastasis [127]. Persistent inflammation in the tumor microenvironment leads to the recruitment of immune cells, such as tumor-associated macrophages (TAMs), myeloid-derived suppressor cells (MDSCs), and regulatory T cells (Tregs), which suppress anti-tumor immune responses and promote immune tolerance [128].

Impaired efferocytosis, the inefficient clearance of apoptotic cells, can contribute to chronic inflammation and immune dysregulation. The accumulation of apoptotic cells leads to secondary necrosis and the release of pro-inflammatory cytokines, chemokines, and damage-associated molecular patterns (DAMPs), which perpetuate inflammation and suppress anti-tumor immunity [129]. Additionally, impaired efferocytosis can lead to the accumulation of apoptotic tumor cells, which fail to induce effective anti-tumor immune responses and may even promote immunosuppression through the release of immunosuppressive factors [129]. Tumor cells employ various strategies to evade immune surveillance and elimination, including the downregulation of major histocompatibility complex (MHC) molecules, the expression of immune checkpoint molecules (e.g., PD-L1, CTLA-4), the induction of immune tolerance, and the recruitment of immunosuppressive cells [125]. These mechanisms inhibit effector T cell function, promote T cell exhaustion, and facilitate immune evasion, allowing tumor cells to evade immune destruction and proliferate unchecked [130]. To counteract tumor immune escape and restore anti-tumor immunity, the following strategies can be proposed:Enhancing Efferocytosis: Promoting efficient efferocytosis can help reduce inflammation and remove immunosuppressive apoptotic cells from the tumor microenvironment. Strategies to enhance efferocytosis include targeting efferocytosis receptors on phagocytes, modulating signaling pathways involved in efferocytosis, and promoting the resolution of inflammation [129].Immune Checkpoint Inhibition: Blocking immune checkpoint molecules, such as PD-1/PD-L1 and CTLA-4, can restore anti-tumor immune responses and enhance T cell-mediated tumor killing. Immune checkpoint inhibitors (ICIs) have shown efficacy in a variety of cancers by unleashing the immune system to recognize and attack tumor cells [131].Targeting Tumor-Associated Inflammation: Modulating the inflammatory microenvironment of tumors can enhance anti-tumor immunity and inhibit tumor growth. This can be achieved through targeting inflammatory mediators, such as cytokines and chemokines, or by repolarizing TAMs and MDSCs from immunosuppressive to anti-tumor phenotypes [132].Immunotherapy: Various immunotherapeutic approaches, including cancer vaccines, adoptive cell therapy (e.g., CAR-T cells), and cytokine therapy, aim to boost anti-tumor immune responses and overcome immune evasion mechanisms employed by tumors. These strategies harness the power of the immune system to recognize and eliminate tumor cells [133].Tumor immune escape is facilitated by chronic inflammation and impaired efferocytosis, which create an immunosuppressive microenvironment that is conducive to tumor growth [126]. Strategies aimed at restoring immunological responses, enhancing efferocytosis, and targeting tumor-associated inflammation can help to counteract tumor immune escape and improve the efficacy of cancer immunotherapy [134].Severe Infectious Diseases: Compromised immune responses, particularly linked to impaired efferocytosis and chronic inflammation, heighten a patient’s susceptibility to severe infections [135]. Apoptotic cell debris in tissues fosters a pathogen-friendly environment when efferocytosis is hindered, leading to the accumulation of dead cells and potential pathogen reservoirs [136]. Chronic inflammation associated with impaired efferocytosis amplifies pro-inflammatory cytokine and chemokine production, weakening overall immune defenses and promoting pathogen survival [135]. Prolonged exposure to inflammation and apoptotic cell debris impairs immune cell function, hindering pathogen recognition and elimination [137]. Continuous exposure to apoptotic cell debris may induce immune tolerance, compromising the immune system’s ability to mount effective defenses against infections [138]. Some pathogens exploit this compromised immune response, utilizing apoptotic cell debris as a protective niche to evade immune surveillance, replicate, and cause secondary infections [139]. This scenario is prominent in conditions featuring chronic inflammation or impaired efferocytosis, such as autoimmune diseases or chronic inflammatory disorders, further perpetuating the cycle [75]. The failure to efficiently clear apoptotic cells contributes to prolonged and dysregulated inflammation, leading to tissue damage, disease progression, and associated symptoms [14]. Enhancing apoptotic cell clearance could offer novel therapeutic approaches for managing these conditions.

## 5. Therapeutic Implications

Comprehending the significance of compromised efferocytosis in chronic inflammation offers potential avenues for therapeutic intervention. Overall, the persistent nature of chronic inflammation resulting from impaired efferocytosis highlights the importance of addressing the clearance of apoptotic cells as a therapeutic strategy [140]. Enhancing efferocytosis may help to regulate the inflammatory response and mitigate the detrimental effects associated with chronic inflammation in various diseases (Table 1) [75]. Researchers are currently exploring various therapeutic implications and strategies in this context.
Enhancement of Efferocytosis: Developing pharmacological agents that are capable of boosting efferocytosis represents a promising therapeutic avenue [141]. This strategy may entail medications designed to enhance the recognition and clearance of apoptotic cells by phagocytes. Exploring the targeted modulation of signaling pathways involved in efferocytosis offers potential for improving its efficiency, including interventions that regulate the interactions between phagocytes and apoptotic cells [142].Anti-Inflammatory Therapies/ Cytokine Modulation: Modulating the levels of pro-inflammatory cytokines released in response to impaired efferocytosis may be considered [143]. Anti-inflammatory therapies targeting cytokines such as interleukin-6 (IL-6) and tumor necrosis factor-alpha (TNF-α) could help dampen chronic inflammation [142,144]. Developing therapeutic strategies that promote the production of resolution-inducing mediators, which help to resolve inflammation, could be beneficial [145].Immunomodulation: Modulating the activity of immune cells involved in the inflammatory response, such as macrophages and T cells, may help restore the balance and prevent excessive inflammation [146]. In autoimmune diseases associated with impaired efferocytosis, therapies aimed at modulating the immune response and preventing the recognition of self-antigens may be explored [147].Lipid Mediators and DAMP Clearance: Specialized Pro-Resolving Lipid Mediators (SPMs) such as resolvins and lipoxins are lipid mediators that actively promote the resolution of inflammation [148]. Developing strategies to enhance the production or administration of SPMs could be a therapeutic approach. Addressing the clearance of damage-associated molecular patterns (DAMPs) released during impaired efferocytosis may be crucial. Strategies that can enhance the removal of these molecules could help to mitigate the inflammatory response [149].Personalized Medicine/Patient-Specific Approaches: Considering the heterogeneity of inflammatory disorders, personalized medicine approaches that consider individual variations in efferocytosis and immune responses may be explored [150]. Identifying biomarkers associated with impaired efferocytosis and heightened inflammatory responses in individual patients can help tailor treatment strategies [151]. Biomarkers may include circulating levels of apoptotic cells, phagocyte function markers, cytokine profiles, and genetic variants associated with efferocytosis and inflammation [152].

Genetic variations can influence efferocytosis efficiency and immune responses. Genome-wide association studies (GWAS) and whole-genome sequencing can help identify genetic variants associated with susceptibility to inflammatory disorders and response to treatment [153]. Genome-wide association studies (GWAS) analyze genetic variations across the entire genome to identify associations between specific genetic variants and traits or diseases [154]. GWAS offer valuable insights into the genetic factors influencing drug response variability, aiding in drug selection and dosing optimization through several key mechanisms:Identifying Drug Targets: GWAS can pinpoint genetic variants linked to disease susceptibility or treatment response, informing the identification of potential drug targets for more targeted therapies [155].Predicting Drug Response: Genetic variations identified through GWAS can affect drug metabolism, pharmacodynamics, and adverse reactions, enabling clinicians to predict individual patients’ responses to drugs and tailor treatment regimens accordingly [156].Personalizing Drug Therapy: Pharmacogenomics utilizes GWAS findings to personalize drug therapy based on patients’ genetic profiles, minimizing adverse effects, and enhancing treatment outcomes by selecting the most suitable drugs and dosages [157].Optimizing Drug Dosing: GWAS uncovers genetic variants influencing drug pharmacokinetics and pharmacodynamics, informing optimal dosing strategies tailored to different patient populations to achieve therapeutic levels effectively [158].Drug Development and Precision Medicine: GWAS results guide drug development by highlighting genetic targets and patient subpopulations that are likely to benefit from novel therapies, aligning with the principles of precision medicine to deliver tailored healthcare interventions [159].

This information can guide personalized treatment selection and dosage optimization. Characterizing the immune cell populations and their functional profiles in individual patients can provide insights into their immune status and potential dysregulation [160,161]. Flow cytometry, mass cytometry, and single-cell RNA sequencing techniques can be used to analyze immune cell subsets and their activity levels, guiding the selection of targeted therapies [162].

Personalized treatment involves targeting the specific molecular pathways involved in efferocytosis and inflammation. Therapies can include enhancing efferocytosis with phagocytic receptor agonists or inhibiting pro-inflammatory signaling pathways, tailored to individual patient characteristics and disease severity [163]. Regular monitoring of disease activity and treatment response using biomarkers, imaging studies, and clinical assessments is crucial for optimizing personalized treatment [164]. This patient-specific approach holds promise for improving the management and outcomes of inflammatory disorders, enhancing therapeutic efficacy, minimizing adverse effects, and ultimately improving patient outcomes [165].

## 6. Mitigation

Mitigating the effects of overnutrition and lipotoxicity involves a multifaceted approach that addresses dietary habits, lifestyle choices, and potential underlying metabolic dysregulations (Table 2) [166]. Below are some potential therapeutic paths that can be used to alleviate the negative consequences of excessive nutrition and lipotoxicity:Nutritional Counseling and Education: This involves working with healthcare professionals, including registered dietitians, to develop personalized dietary plans based on individual needs, preferences, and health conditions, implementing controlled caloric intake to achieve and maintain a healthy weight [167], and emphasizing a balanced distribution of macronutrients (carbohydrates, proteins, and fats) to meet nutritional needs without an excessive caloric intake [168]. The concept of energy balance refers to the relationship between energy intake (from food and beverages) and energy expenditure (through physical activity, metabolic processes, and other bodily functions) [169]. When the energy intake and energy expenditure are equal, there is said to be an energy balance. When the energy intake is greater than the energy expenditure, a positive energy balance occurs, resulting in weight gain over time. Conversely, when the energy expenditure is greater than the energy intake, a negative energy balance occurs, resulting in weight loss over time [170] (Figure 4).Regular Physical Activity: This involves incorporating regular physical activity tailored to individual fitness levels and health conditions and engaging in aerobic exercises to improve cardiovascular health and enhance metabolic function, including the use of resistance training to build muscle mass, which can contribute to improved insulin sensitivity [171].Weight Management In cases of severe obesity or metabolic disorders, medical supervision may be required, and interventions like bariatric surgery could be contemplated [172].Lipid-Lowering Medications: This involves administering statin medications to decrease elevated cholesterol levels and mitigate the risk of cardiovascular events, as well as using fibrates to target triglyceride levels and improve lipid profiles and incorporating omega-3 fatty acid supplements, which may have beneficial effects on lipid metabolism [173].Insulin Sensitizers: This involves administering metformin, an insulin-sensitizing medication commonly used in the management of type 2 diabetes. In certain cases, thiazolidinediones may be considered to improve insulin sensitivity [174].Anti-inflammatory Agents: This involves implementing anti-inflammatory lifestyle choices, including a diet rich in anti-inflammatory foods (e.g., fruits, vegetables, and fatty fish) and regular exercise. In some cases, medications with anti-inflammatory properties may be considered to address inflammation associated with lipotoxicity [175].Monitoring and Screening: Periodic health check-ups can be carried out to monitor metabolic parameters, lipid profiles, and other indicators of metabolic health. Screening for comorbidities can also help in identifying and managing comorbid conditions such as diabetes, hypertension, and cardiovascular diseases [176].Psychosocial Support: A comprehensive approach to combatting overnutrition includes behavioral counseling, lifestyle coaching, and public health initiatives [177]. Behavioral counseling provides crucial support and addresses emotional or psychological factors contributing to overnutrition. Lifestyle coaching guides individuals in adopting sustainable changes to their dietary and activity habits for long-term well-being [178].

## 7. Conclusions

In conclusion, addressing the challenges posed by overnutrition and lipotoxicity requires a holistic and personalized approach to promote metabolic health and reduce associated risks. The consequences of overnutrition, characterized by excessive caloric intake, and lipotoxicity, marked by the accumulation of lipids leading to cellular dysfunction, are complex and can contribute to a range of metabolic disorders, cardiovascular diseases, and other health complications.

Key therapeutic avenues to mitigate the effects of overnutrition and lipotoxicity include nutritional counseling, lifestyle modifications, weight management strategies, and the judicious use of medications targeting lipid levels and insulin sensitivity. Regular physical activity, balanced dietary choices, and interventions to reduce inflammation are essential components of a comprehensive strategy. Individualized care, considering factors such as genetics, pre-existing health conditions, and metabolic status, is crucial in developing effective interventions. Regular monitoring, health check-ups, and screening for comorbidities contribute to the early detection and management of metabolic issues.

Furthermore, public health initiatives and policy changes play a vital role in creating environments that support healthier lifestyles. Education, community programs, and policies promoting access to nutritious foods and opportunities for physical activity are essential for preventing and managing the consequences of overnutrition and lipotoxicity at the population level. Collaboration between individuals, healthcare professionals, public health agencies, and communities is paramount in navigating the complexities of overnutrition and lipotoxicity. By adopting a proactive and integrative approach, it is possible to reduce the prevalence of metabolic disorders, improve overall health outcomes, and enhance the well-being of individuals affected by these conditions.

## Figures and Tables

**Figure 1 biology-13-00241-f001:**
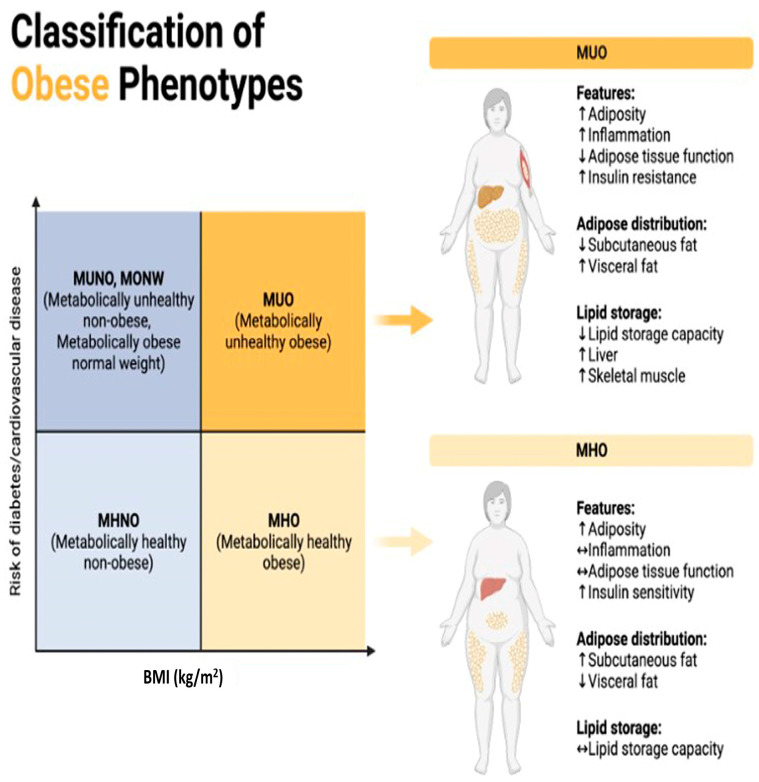
Classification of obese phenotypes. Figure 1 depicts the classification of obese phenotypes based on Body Mass Index (BMI) and the associated risk of developing diabetes and cardiovascular disease (CVD). The x-axis represents BMI categories, while the y-axis represents the risk of diabetes and CVD. The figure is divided into four quadrants, each representing a distinct obese phenotype. MUNO)/MONW: Despite having a normal BMI (<25 kg/m^2^), individuals in this quadrant display metabolic abnormalities akin to obesity, including insulin resistance, dyslipidemia, and hypertension, elevating their risk of diabetes and cardiovascular disease comparable to or even exceeding that of obese individuals. MUO: With a high BMI (≥30 kg/m^2^) and concurrent metabolic abnormalities like insulin resistance, dyslipidemia, and hypertension, individuals in this quadrant face the highest risk of developing diabetes and cardiovascular disease among all obese phenotypes due to the combination of obesity and metabolic dysfunction. MHNO: Individuals here possess a normal BMI (<25 kg/m^2^) and lack metabolic abnormalities associated with obesity, indicating a lower risk of diabetes and cardiovascular disease compared to their metabolically unhealthy obese counterparts. MHO: Despite having a high BMI (≥30 kg/m^2^), individuals in this quadrant do not exhibit metabolic abnormalities linked to obesity, suggesting a relatively lower risk of diabetes and cardiovascular disease compared to metabolically unhealthy obese individuals, which is indicative of a more favorable metabolic profile despite excess weight.

**Figure 2 biology-13-00241-f002:**
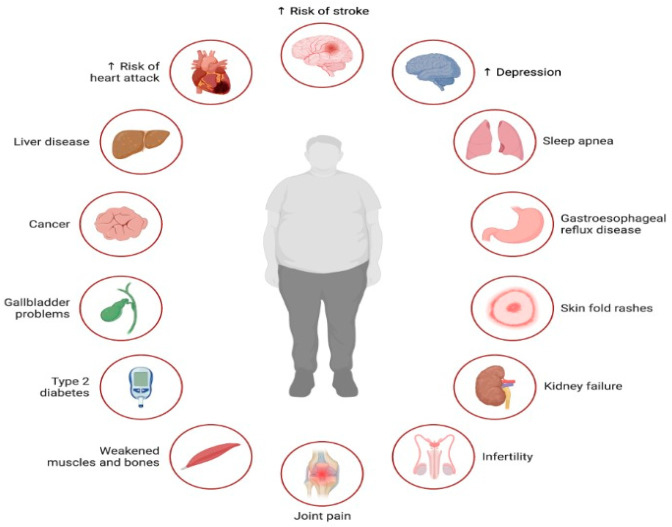
Illustration of the multifaceted impact of lipotoxicity, characterized by excessive lipid accumulation in non-adipose tissues, on various aspects of health. Lipotoxicity contributes to the development or exacerbation of several health conditions across different organ systems. Lipotoxicity exerts a profound impact on health, contributing to the development or exacerbation of numerous health conditions, including cardiovascular disease, liver disease, cancer, diabetes, musculoskeletal disorders, infertility, mental health disorders, and sleep-related breathing disorders.

**Figure 3 biology-13-00241-f003:**
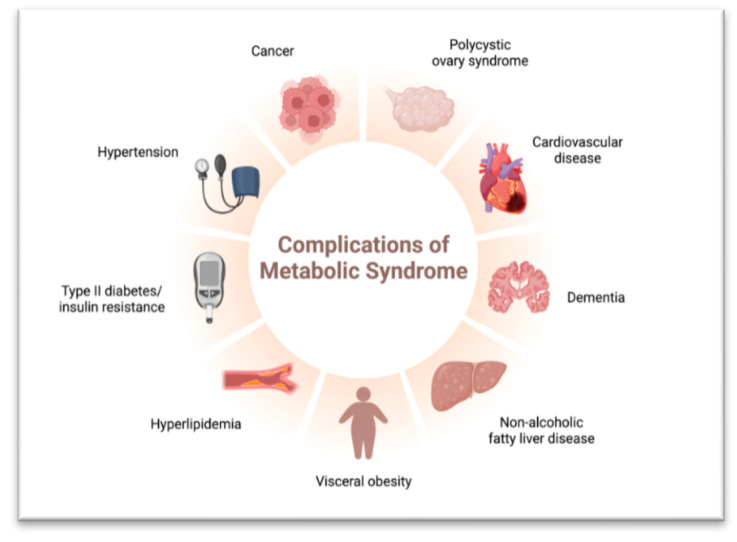
Complications of Metabolic Syndrome.

**Figure 4 biology-13-00241-f004:**
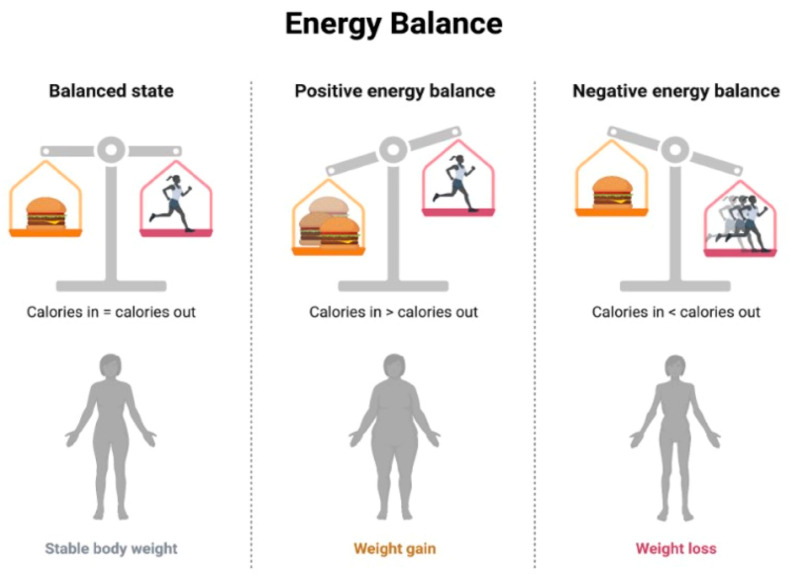
Energy Balance.

**Table 1 biology-13-00241-t001:** Different approaches for therapeutic intervention targeting efferocytosis.

Approach	Description	Personalized Aspect
Pharmacological Modulation	Development of pharmacological agents targeting efferocytosis receptors or signaling pathways involved in the recognition and engulfment of apoptotic cells.	Tailoring pharmacological interventions based on individual patient characteristics, such as genetic variations affecting efferocytosis receptors or underlying diseases.
Immunomodulatory Therapy	Utilization of immunomodulatory agents, such as cytokines, growth factors, or monoclonal antibodies, to enhance efferocytosis activity of immune cells.	Customizing immunomodulatory therapies based on the immune profile and responsiveness of individual patients, including considerations for immune cell function and cytokine levels.
Stem Cell Therapy	Administration of stem cells, such as mesenchymal stem cells (MSCs), which possess immunomodulatory properties and can promote efferocytosis by enhancing phagocytic activity of macrophages.	Selecting optimal stem cell sources and dosages based on individual patient characteristics, including age, underlying diseases, and immune status.
Gene Therapy	Genetic manipulation of efferocytosis-related genes or pathways using gene editing technologies, viral vectors, or RNA interference to enhance phagocytic capacity of immune cells.	Targeting specific genetic mutations or polymorphisms associated with impaired efferocytosis in individual patients and designing personalized gene therapy approaches.
Nutritional Interventions	Dietary interventions targeting nutrient deficiencies or imbalances that may impair efferocytosis function, such as omega-3 fatty acids, antioxidants, or vitamins.	Designing personalized nutrition plans based on individual dietary habits, nutritional status, and metabolic needs to optimize efferocytosis efficiency and overall immune function.
Lifestyle Modifications	Adoption of lifestyle habits, such as regular physical activity, stress management, and smoking cessation, which can positively influence immune function and efferocytosis activity.	Tailoring lifestyle recommendations to accommodate individual preferences, capabilities, and socioeconomic factors.
Combination Therapies	Integration of multiple therapeutic modalities to synergistically enhance efferocytosis and overall immune function.	Developing personalized combination therapy regimens based on individual patient profiles, including disease severity, treatment response, and potential drug interactions.

**Table 2 biology-13-00241-t002:** Various approaches to mitigate the risks associated with overnutrition and lipotoxicity.

Approach	Description
Dietary Modifications	Emphasize a balanced diet rich in fruits, vegetables, whole grains, lean proteins, and healthy fats. Reduce intake of processed foods, sugary beverages, saturated fats, and trans fats. Monitor portion sizes and caloric intake.
Regular Physical Activity	Engage in regular aerobic exercise, strength training, and flexibility exercises. Aim for at least 150 min of moderate-intensity or 75 min of vigorous-intensity exercise per week. Incorporate physical activity into daily routines.
Weight Management	Maintain a healthy body weight through a combination of diet, exercise, and lifestyle modifications. Set realistic weight loss goals and seek support from healthcare professionals or support groups if needed.
Pharmacotherapy	Consider pharmacological interventions, such as anti-obesity medications or lipid-lowering drugs, under the guidance of a healthcare provider. May be recommended for individuals with obesity or dyslipidemia.
Bariatric Surgery	Surgical procedures such as gastric bypass, sleeve gastrectomy, or gastric banding may be considered for individuals with severe obesity or obesity-related comorbidities who have not responded to other weight loss interventions.
Nutritional Supplements	Consider supplementation with vitamins, minerals, and omega-3 fatty acids to address nutrient deficiencies and support overall health. Consult with healthcare provider or registered dietitian for personalized guidance.
Lifestyle Modifications	Adopt healthy lifestyle habits such as getting adequate sleep, managing stress, and avoiding smoking and excessive alcohol consumption. Practice mindful eating, focusing on hunger cues, portion control, and enjoyment of food.
Medical Monitoring	Regularly monitor blood glucose levels, lipid profiles, blood pressure, and other metabolic parameters to assess health status and track progress. Schedule routine check-ups with healthcare providers for personalized guidance.

## Data Availability

The original contributions presented in the study are included in the article, further inquiries can be directed to the corresponding author/s.

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
