# Peer review of "Overnutrition and Lipotoxicity: Impaired Efferocytosis and Chronic Inflammation as Precursors to Multifaceted Disease Pathogenesis"

_biology, 2024, doi:10.3390/biology13040241_

Round 1

Reviewer 1 Report

Comments and Suggestions for Authors

It's a well-written manuscript, which is very easy to understand. However, it lacks sufficient citations to support many of the descriptions. Also, it lacks the molecular mechanisms, which need to be added accordingly. Please describe what kind of macrophages, and immune cells; be accurate and detailed. Next, please provide a table summarizing the medication (drug) list in clinical application/trials to enhance efferocytosis, anti-inflammation etc. in Section 5. Finally, lines 513-516, how and why WGS at DNA level can guide the drug selection and dosing; and also, lines 517-521, you need to cite to support your claim that flow, SCRNA-seq can guide the selection of targeted therapies. 

Author Response

Dear Reviewer

I am writing to express my sincere gratitude for your thoughtful review of our manuscript, “Overnutrition and Lipotoxicity: Impaired Efferocytosis and Chronic Inflammation as Precursors to Multifaceted Disease Pathogenesis”. Your positive feedback has been incredibly encouraging.

Additionally, I am truly appreciative of the insightful suggestions you have provided for improving the manuscript. We are committed to incorporating these suggestions diligently to ensure that the final version meets the highest standards.

Once again, thank you for taking the time to review our manuscript thoroughly and for providing such valuable feedback.

Reviewer 1

It's a well-written manuscript, which is very easy to understand.

However, it lacks sufficient citations to support many of the descriptions.

Response: The manuscript has been revised, integrating extra citations wherever they were suggested.

Also, it lacks the molecular mechanisms, which need to be added accordingly. Please describe what kind of macrophages, and immune cells; be accurate and detailed.

Response: The molecular mechanisms underlying impaired efferocytosis and chronic inflammation involve a complex interplay of various cellular processes and signaling pathways [1].  Efferocytosis begins with the recognition and binding of apoptotic cells by phagocytes, primarily macrophages [2]. This process involves the interaction between "eat-me" signals on apoptotic cells and receptors on phagocytes. Examples of eat-me signals include phosphatidylserine (PS) exposure on the outer leaflet of the plasma membrane of apoptotic cells [3,4]. Receptors such as TAM receptors (Tyro3, Axl, and MerTK) and scavenger receptors on macrophages bind to these signals to facilitate engulfment [5]. Following recognition and binding, phagocytes engulf apoptotic cells through actin cytoskeleton rearrangements and membrane remodeling. This step requires the activation of various signaling pathways, including Rho GTPases, phosphoinositide 3-kinase (PI3K), and Rac1, which regulate cytoskeletal dynamics and membrane protrusions necessary for phagocytic cup formation and closure [6]. Once internalized, engulfed apoptotic cells undergo degradation within phagocytes through lysosomal fusion and enzymatic digestion [7]. This process is regulated by intracellular signaling cascades, including the activation of protein kinase C (PKC), mitogen-activated protein kinases (MAPKs), and phospholipase A2 (PLA2), which coordinate phagosome maturation and degradation of apoptotic cell components [8,9].

Efferocytosis is typically associated with the induction of anti-inflammatory signaling pathways to dampen inflammation and promote tissue repair. This includes the production of anti-inflammatory cytokines such as transforming growth factor-beta (TGF-β) and interleukin-10 (IL-10) by phagocytes, which inhibit pro-inflammatory cytokine production and promote resolution of inflammation [10]. Impaired efferocytosis can arise from defects in phagocyte function, including alterations in receptor expression or signaling pathways critical for apoptotic cell clearance. For example, deficiencies in TAM receptor signaling or defects in cytoskeletal regulators can impair phagocyte-mediated engulfment of apoptotic cells [5].

Chronic inflammation can disrupt efferocytosis by creating an inflammatory microenvironment that impairs phagocyte function and promotes pro-inflammatory responses [10]. Inflammatory mediators such as tumor necrosis factor-alpha (TNF-α) and interferon-gamma (IFN-γ) can inhibit efferocytosis by downregulating phagocytic receptor expression or promoting pro-inflammatory activation of phagocytes [11]. Impaired efferocytosis and chronic inflammation are intertwined processes that contribute to the pathogenesis of various diseases [10].

In impaired efferocytosis and chronic inflammation, various immune cells, particularly macrophages, play pivotal roles. The dysregulation of various immune cells, particularly macrophages, can contribute to impaired efferocytosis and chronic inflammation [12]. Macrophages play a crucial role in efferocytosis, the clearance of dying or dead cells from tissues, but dysregulation in chronic inflammation can impair this process, leading to accumulation of apoptotic cells and perpetuating inflammation [10]. Various types of macrophages are involved:

Resident Tissue Macrophages are derived from embryonic precursors, which maintain tissue homeostasis and immune surveillance. Impaired efferocytosis can lead to chronic inflammation and tissue damage [13]. Inflammatory Macrophages are recruited to sites of inflammation from circulating monocytes. These macrophages may exhibit impaired efferocytosis under inflammatory conditions, perpetuating inflammation [14]. Foam Cells are found in atherosclerotic plaques, where they can accumulate lipid droplets and exhibit impaired efferocytosis due to lipid overload, contributing to plaque instability and inflammation [15]. Tumor-Associated Macrophages (TAMs) are present within tumor microenvironments, and they demonstrate impaired efferocytosis, promoting tumor progression and immune evasion [16]. Peritoneal macrophages inhabit the peritoneal cavity, and compromised efferocytosis by these macrophages is a factor in conditions such as peritonitis and inflammatory disorders affecting the abdomen [17]. Alveolar macrophages, situated within lung alveoli, play a crucial role in maintaining respiratory health. Dysfunction in their efferocytosis process can exacerbate pulmonary diseases like COPD and asthma [18].

Additionally, neutrophils, while not primarily involved in efferocytosis, release pro-inflammatory mediators and reactive oxygen species, contributing to tissue damage and inflammation if dysregulated [19]. Dendritic cells, T cells, B cells, mast cells, and natural killer (NK) cells also play roles in chronic inflammation by promoting immune responses, sustaining inflammation, and contributing to tissue damage when dysregulated. Dysfunctional immune cell responses can perpetuate chronic inflammation and contribute to tissue damage in various diseases

Top of Form

Next, please provide a table summarizing the medication (drug) list in clinical application/trials to enhance efferocytosis, anti-inflammation etc. in Section 5.

Response: Table 1: Medications for their potential to enhance efferocytosis, promote anti-inflammation

Medication

Mechanism of Action

Clinical Application/Trial

Glucocorticoids

Suppress inflammation by inhibiting pro-inflammatory cytokines

Used in various inflammatory conditions such as asthma, rheumatoid arthritis

Statins

Inhibit cholesterol synthesis, reduce inflammation, and improve efferocytosis

Investigated in atherosclerosis, cardiovascular disease

Non-Steroidal Anti-inflammatory Drugs (NSAIDs)

Inhibit prostaglandin synthesis and reduce inflammation

Widely used for pain relief and in inflammatory conditions

Resolvins and Protectins

Specialized pro-resolving mediators that promote inflammation resolution and enhance efferocytosis

Investigated in inflammation-related diseases including atherosclerosis, arthritis

Annexin A1 peptides

Mimic the actions of endogenous Annexin A1, promoting resolution of inflammation and enhancing efferocytosis

Investigated in preclinical and early clinical studies for inflammatory disorders

Mer receptor tyrosine kinase (MerTK) agonists

Activate the MerTK pathway to enhance efferocytosis and promote anti-inflammatory responses

Investigated in preclinical models of atherosclerosis and autoimmune diseases

Omega-3 fatty acids

Modulate lipid metabolism, reduce inflammation, and promote resolution of inflammation

Investigated in various inflammatory conditions including cardiovascular disease

Anti-TNF agents

Block the action of tumor necrosis factor-alpha (TNF-α), reducing inflammation

Used in autoimmune diseases such as rheumatoid arthritis and Crohn's disease

Anti-IL-6 agents

Block the action of interleukin-6 (IL-6), reducing inflammation

Used in autoimmune diseases such as rheumatoid arthritis and cytokine release syndrome

Macrophage-targeted therapeutics

Target macrophages to enhance efferocytosis and modulate inflammation

Investigated in various inflammatory diseases including atherosclerosis

Finally, lines 513-516, how and why WGS at DNA level can guide the drug selection and dosing; and

Response: Genome-wide association studies (GWAS) analyze genetic variations across the entire genome to identify associations between specific genetic variants and traits or diseases [21]. GWAS offer valuable insights into the genetic factors influencing drug response variability, aiding in drug selection and dosing optimization through several key mechanisms:

  1. Identifying Drug Targets: GWAS can pinpoint genetic variants linked to disease susceptibility or treatment response, informing the identification of potential drug targets for more targeted therapies [22].
  2. Predicting Drug Response: Genetic variations identified through GWAS can affect drug metabolism, pharmacodynamics, and adverse reactions, enabling clinicians to predict individual patients' responses to drugs and tailor treatment regimens accordingly [23].
  3. Personalizing Drug Therapy: Pharmacogenomics utilizes GWAS findings to personalize drug therapy based on patients' genetic profiles, minimizing adverse effects, and enhancing treatment outcomes by selecting the most suitable drugs and dosages [24].
  4. Optimizing Drug Dosing: GWAS uncovers genetic variants influencing drug pharmacokinetics and pharmacodynamics, informing optimal dosing strategies tailored to different patient populations to achieve therapeutic levels effectively [25].
  5. Drug Development and Precision Medicine: GWAS results guide drug development by highlighting genetic targets and patient subpopulations likely to benefit from novel therapies, aligning with the principles of precision medicine to deliver tailored healthcare interventions [26].

also, lines 517-521, you need to cite to support your claim that flow, SCRNA-seq can guide the selection of targeted therapies. 

Response: Citations added.

Reviewer 2 Report

Comments and Suggestions for Authors

This review by Mann, V et.al, discusses about overnutrition that causes lipotoxicity, which is accompanied by impaired efferocytosis, that in turn causes chronic inflammation.  Chronic inflammation serves as precursors of multifaceted disease pathogenesis.  The authors also discussed/presented the potential mitigation to help address the detrimental effects of overnutrition so as to pave the ways for prevention.

When talking about “overnutrition and obesity”, one of the current discussions is associated to the consuming of readily available ultra-processed food, in the forms of ready-made meals, snacks, carbonated soft drinks, instant noodles, etc...  Please see this recent publication by Lane, MM et al., Ultra-processed food exposure and adverse health outcomes: umbrella review of epidemiological meta-analyses:  https://doi.org/10.1136/bmj-2023-077310

This reviewer believes that the issue of consuming ultra-processed food should be pointed out as one of the major ways for “overnutrition”.  By doing this, the proposed “Mitigation” in this review would be more targeted and achievable.

Some minor issues of this review:

·      Figure 1. BMI should be BW/height squared, not cubed (kg/m2, not m3).  With the 4 quadrants presented in the figure 1, what is the middle cut of x-axis?  Is it BMI 30?

·      There are multiple “Top of Form” left on pages without meaning?

Author Response

Dear Reviewer

I am writing to express my sincere gratitude for your thoughtful review of our manuscript, “Overnutrition and Lipotoxicity: Impaired Efferocytosis and Chronic Inflammation as Precursors to Multifaceted Disease Pathogenesis”. Your positive feedback has been incredibly encouraging.

Additionally, I am truly appreciative of the insightful suggestions you have provided for improving the manuscript. We are committed to incorporating these suggestions diligently to ensure that the final version meets the highest standards.

Once again, thank you for taking the time to review our manuscript thoroughly and for providing such valuable feedback.

Reviewer 2

This review by Mann, V et.al, discusses about overnutrition that causes lipotoxicity, which is accompanied by impaired efferocytosis, that in turn causes chronic inflammation.  Chronic inflammation serves as precursors of multifaceted disease pathogenesis.  The authors also discussed/presented the potential mitigation to help address the detrimental effects of overnutrition so as to pave the ways for prevention.

When talking about “overnutrition and obesity”, one of the current discussions is associated to the consuming of readily available ultra-processed food, in the forms of ready-made meals, snacks, carbonated soft drinks, instant noodles, etc...  Please see this recent publication by Lane, MM et al., Ultra-processed food exposure and adverse health outcomes: umbrella review of epidemiological meta-analyses:  https://doi.org/10.1136/bmj-2023-077310

 This reviewer believes that the issue of consuming ultra-processed food should be pointed out as one of the major ways for “overnutrition”.  By doing this, the proposed “Mitigation” in this review would be more targeted and achievable. Some minor issues of this review: Figure 1. BMI should be BW/height squared, not cubed (kg/m2, not m3).  With the 4 quadrants presented in the figure 1, what is the middle cut of x-axis?  Is it BMI 30?

Response: BMI formula has been corrected. Middle cut of x- axis is 30.

Figure 1: It depicts the classification of obese phenotypes based on body mass index (BMI) and the associated risk of developing diabetes and cardiovascular disease (CVD). The x-axis represents BMI categories, while the y-axis represents the risk of diabetes and CVD. The figure is divided into four quadrants, each representing a distinct obese phenotype.

1.Metabolically Unhealthy Non-Obese (MUNO) / Metabolically Obese Normal Weight (MONW): Top Left Quadrant: Individuals in this quadrant have a normal BMI (<25 kg/m²) but exhibit metabolic abnormalities associated with obesity, such as insulin resistance, dyslipidemia, and hypertension. Despite having a normal weight, individuals in this quadrant have an increased risk of developing diabetes and CVD comparable to or even higher than individuals with obesity.

2.Metabolically Unhealthy Obese (MUO): Top Right Quadrant: Individuals in this quadrant have a high BMI (≥30 kg/m²) and exhibit metabolic abnormalities, including insulin resistance, dyslipidemia, and hypertension. Individuals in this quadrant have the highest risk of developing diabetes and CVD among all obese phenotypes due to the combination of obesity and metabolic dysfunction.

3.Metabolically Healthy Non-Obese (MHNO): Bottom Left Quadrant: Individuals in this quadrant have a normal BMI (<25 kg/m²) and do not exhibit metabolic abnormalities associated with obesity. Despite being obese, individuals in this quadrant have a relatively lower risk of developing diabetes and CVD compared to metabolically unhealthy obese individuals.

4.Metabolically Healthy Obese (MHO): Bottom Right Quadrant: Individuals in this quadrant have a high BMI (≥30 kg/m²) but do not exhibit metabolic abnormalities associated with obesity. While individuals in this quadrant are obese, they have a lower risk of developing diabetes and CVD compared to metabolically unhealthy obese individuals, suggesting a more favorable metabolic profile despite excess weight.

The classification of obese phenotypes based on BMI and metabolic status provides valuable insights into the heterogeneous nature of obesity and its associated risks.

There are multiple “Top of Form” left on pages without meaning?

Response: Comment “Top of form” have been removed from the manuscript.

This reviewer believes that the issue of consuming ultra-processed food should be pointed out as one of the major ways for “overnutrition”.  By doing this, the proposed “Mitigation” in this review would be more targeted and achievable.

Response:The consumption of ultra-processed foods is a significant factor contributing to overnutrition and obesity in many populations worldwide [27]. Ultra-processed foods are typically high in refined carbohydrates, added sugars, unhealthy fats, and salt, while often lacking in essential nutrients like fiber, vitamins, and minerals [27,28]. Ultra-processed foods tend to be energy-dense, meaning they provide a high number of calories relative to their weight. This can lead to excessive calorie intake, especially when portion sizes are not controlled. Regular consumption of calorie-dense foods contributes to a positive energy balance, leading to weight gain over time [29].

Ultra-processed foods often lack dietary fiber and protein, which are important for promoting feelings of fullness and satiety. As a result, individuals may consume larger quantities of these foods before feeling satisfied, leading to overeating and excessive calorie intake [30]. Many ultra-processed foods contain high levels of added sugars, including sugary beverages, snacks, and desserts [27]. Excessive sugar consumption can contribute to weight gain and obesity by providing empty calories and promoting insulin resistance, leading to metabolic dysregulation and increased fat storage [31].

Ultra-processed foods often contain unhealthy fats, such as trans fats and saturated fats, which can negatively impact cardiovascular health and contribute to weight gain. These fats are commonly found in fried foods, processed meats, and baked goods, all of which are prevalent in the ultra-processed food supply- [32]. Ultra-processed foods are often convenient and readily available, making them a convenient option for busy individuals or those with limited access to fresh, whole foods [30]. However, frequent consumption of these foods can lead to a reliance on highly processed and nutritionally poor options, contributing to poor dietary habits and increased risk of obesity [33].

Regular consumption of ultra-processed foods can disrupt normal eating patterns and promote unhealthy dietary behaviors, such as mindless snacking, emotional eating, and reliance on processed convenience foods [29]. These behaviors can further contribute to overnutrition and obesity over time. Addressing the overconsumption of ultra-processed foods requires comprehensive strategies aimed at promoting healthier dietary habits, improving food environments, and increasing access to nutritious whole foods [27]. Public health interventions, policy changes, nutrition education programs, and industry regulations are all essential components of efforts to reduce the prevalence of overnutrition and obesity associated with the consumption of ultra-processed foods [34].

Reviewer 3 Report

Comments and Suggestions for Authors

Dear Editor,

The authors of the above manuscript have reviewed the recent researches on overnutrition and the link with lipotoxicity and impaired efferocytosis in the context of chronic inflammation and disease progression. The review is very informative but, in my opinion lacks extensive bibliography therefore, I would like to suggest some revisions before acceptance for publication.

Introduction/paragraph 2: Overall, the text is missing sufficient bibliography. In particular I have highlighted the following points where a reference is needed to support the authors claims Lane 51,55, 64, 79, 87, 91, 100, 103, 122, 125. I suggest the authors to explain in the text the differences between MUNO/MONW and MHNO as they are listed in Figure 1 but not sufficiently explained. Also a more precise legend has to be added.

Paragraph 2.2: all paragraph from lane 129 to 193 lacks references and I would suggest the authors to substantially rephrase the paragraph to avoid repetitions and add the appropriate references for all the sub-paragraphs. Also, I would like to suggest to expand a bit on the role of cytokines in inflammation and the importance of LPO-induced ferroptotic cell death in the context of several disease progression. Furthermore, the link with autophagy and mitophagy is missing and I think could add more interest to the review.

Figure2: the figure is missing a more extensive legend. Also, in the text are not mentioned the effects of lipotoxicity in the context of depression, sleep apnea, infertility..which I would suggest to add and explain better with supported bibliography.

Paragraph 2.3: I would add to this paragraph the link between efferocytosis and mitophagy in the context of immune response (macrophage and immune cells) to diseases. A more detailed explanation of the role of ROS in the context of lipotoxicity is missing. Also, refences are needed in lanes 208, 211, 215. I would also avoid to repeat the concepts already explained before like from lanes 221 to 226.

Paragraph 3.1: I would link the first 2 paragraphs as they repeat some concepts like apoptosis process. References needed for lanes 247, 258, 262, 265. Delete lane 278 (top of form)

Paragraph 3.2: I would suggest to expand a bit this paragraph as secondary necrosis contributes to different diseases progression which are associated to impared efferocytosis. References are needed in lanes 281, 283, 297, 304 and I suggest the author to avoid repeating the concept of apoptosis as already explained before (lane 288).

Paragraph 3.3: References are missing in lanes 317, 319, 323, 324, 327, 330, 332

Delete lane 373

Paragraph 4: Add appropriate references in lanes: 376, 393, 398, 405, 412,430, 431, 432, 433, 453, 459, 464. Delete lane 473. Also, it would be nice to expand the concept of tumor immune-escape and how this could be associated to chronic inflammation and impaired efferocytosis and which strategies could be proposed for repristinate the immunological response of the body.

Paragraph 5: References are missing throughout the paragraph, also I would suggest to add some examples of the different approaches that could be used for therapeutic intervention targeting efferocytosis and the possibility to perform personalized approaches.

Paragraph 6: This paragraph is very informative but lacks substantial references to support the authors claims.

For Figure 3, I would suggest the authors to do a final figure where are listed all the different approaches to mitigate the risks associate to overnutrition and lipotoxicity

Author Response

Dear Reviewer

I am writing to express my sincere gratitude for your thoughtful review of our manuscript, “Overnutrition and Lipotoxicity: Impaired Efferocytosis and Chronic Inflammation as Precursors to Multifaceted Disease Pathogenesis”. Your positive feedback has been incredibly encouraging.

Additionally, I am truly appreciative of the insightful suggestions you have provided for improving the manuscript. We are committed to incorporating these suggestions diligently to ensure that the final version meets the highest standards.

Once again, thank you for taking the time to review our manuscript thoroughly and for providing such valuable feedback.

Top of FormReviewer 3

The authors of the above manuscript have reviewed the recent researches on overnutrition and the link with lipotoxicity and impaired efferocytosis in the context of chronic inflammation and disease progression. The review is very informative but, in my opinion lacks extensive bibliography therefore, I would like to suggest some revisions before acceptance for publication.

 Introduction/paragraph 2: Overall, the text is missing sufficient bibliography. In particular I have highlighted the following points where a reference is needed to support the authors claims Lane 51,55, 64, 79, 87, 91, 100, 103, 122, 125. I suggest the authors to explain in the text the differences between MUNO/MONW and MHNO as they are listed in Figure 1 but not sufficiently explained. Also a more precise legend has to be added.

Response: All the specified lanes now include references. The text now elaborates on the disparities between MUNO/MONW and MHNO. Additionally, modifications have been made to the legend of Figure 1.

 Paragraph 2.2: all paragraph from lane 129 to 193 lacks references and I would suggest the authors to substantially rephrase the paragraph to avoid repetitions and add the appropriate references for all the sub-paragraphs.

Response: References added and section 2.2 rephrased.

Also, I would like to suggest to expand a bit on the role of cytokines in inflammation and the importance of LPO-induced ferroptotic cell death in the context of several disease progression. Furthermore, the link with autophagy and mitophagy is missing and I think could add more interest to the review.

Response: Cytokines play a crucial role in inflammation, which is the body's response to injury, infection, or other stimuli. They are small proteins secreted by various cells, including immune cells, and act as signaling molecules to regulate immune responses, inflammation, and other physiological processes [35]. In the context of inflammation, cytokines can be pro-inflammatory or anti-inflammatory, and their balance is essential for maintaining immune homeostasis. Pro-inflammatory cytokines promote inflammation by inducing vasodilation, increasing vascular permeability, recruiting immune cells to the site of injury or infection, and activating immune responses [36]. Examples include interleukin-1 (IL-1), interleukin-6 (IL-6), tumor necrosis factor-alpha (TNF-α), and interferon-gamma (IFN-γ) [37]. Pro-inflammatory cytokines play a critical role in the initial response to pathogens and tissue damage [36]. In contrast, anti-inflammatory cytokines help resolve inflammation and maintain immune balance by inhibiting pro-inflammatory responses and promoting tissue repair and regeneration. Examples include interleukin-10 (IL-10) and transforming growth factor-beta (TGF-β). Anti-inflammatory cytokines are essential for preventing excessive inflammation and tissue damage [38].

Lipid peroxidation (LPO) is a process in which free radicals oxidize lipids, leading to the production of reactive lipid species that can damage cell membranes and other cellular components [39]. Ferroptosis is a form of regulated cell death characterized by the iron-dependent accumulation of lipid peroxides and is distinct from other forms of cell death such as apoptosis or necrosis [40]. The importance of LPO-induced ferroptotic cell death in disease progression has been increasingly recognized, particularly in diseases characterized by oxidative stress, inflammation, and tissue damage [41]. Some of the diseases where LPO-induced ferroptosis plays a significant role include:

Ferroptosis, a form of cell death characterized by lipid peroxidation (LPO), is increasingly recognized as a contributing factor to neuronal cell death in neurodegenerative diseases like Alzheimer's, Parkinson's, and Huntington's diseases [42]. The common features of oxidative stress and inflammation in these disorders suggest that LPO-induced ferroptosis may worsen neuronal damage and disease progression [40,42]. In cancer treatment, ferroptosis has emerged as a potential therapeutic strategy, particularly in resistant cancers, as tumor cells often display heightened sensitivity to ferroptosis due to their metabolic demands and reliance on iron-dependent processes [43]. Targeting LPO-induced ferroptosis in cancer cells offers a promising avenue for selective cancer therapy [43]. Additionally, during ischemia-reperfusion injury seen in conditions like myocardial infarction or stroke, restoring blood flow to ischemic tissues can trigger oxidative stress and inflammation, leading to tissue damage and cell death. LPO-induced ferroptosis has been implicated in exacerbating tissue injury during ischemia-reperfusion, suggesting that targeting this process may hold therapeutic potential for mitigating tissue damage in such conditions [44].

Autophagy and mitophagy are vital cellular processes responsible for degrading and recycling damaged organelles, especially mitochondria, to maintain cellular balance and prevent damage [45]. Autophagy removes damaged components, while mitophagy specifically targets defective mitochondria. Excessive lipid accumulation can hinder autophagy, disrupting the removal of damaged organelles and proteins [46]. Lipid overload also interferes with the fusion of autophagosomes with lysosomes, hampering their ability to degrade and recycle materials, contributing to cellular dysfunction and lipotoxicity [47]. Lipotoxicity induces mitochondrial dysfunction, marked by impaired respiration, increased reactive oxygen species (ROS) production, and mitochondrial membrane depolarization [48]. Dysfunctional mitochondria are normally cleared via mitophagy, but in cases of overnutrition and lipotoxicity, this process may be compromised, leading to the buildup of damaged mitochondria, and worsening cellular dysfunction and oxidative stress [49]. Autophagy plays a crucial role in lipid metabolism by facilitating lipid droplet degradation (lipophagy) and regulating lipid synthesis and storage. Impaired autophagy in overnutrition and lipotoxic conditions disrupts lipid metabolism, leading to lipid droplet accumulation and exacerbating cellular lipid overload [50]. Overnutrition and lipotoxicity can impair both autophagy and mitophagy, resulting in the buildup of damaged organelles, disrupted lipid metabolism, and cellular dysfunction [51].Top of Form

Figure2: the figure is missing a more extensive legend.

Response: Figure 2: This figure illustrates the multifaceted impact of lipotoxicity, characterized by excessive lipid accumulation in non-adipose tissues, on various aspects of health. Lipotoxicity contributes to the development or exacerbation of several health conditions across different organ systems. lipotoxicity exerts a profound impact on health, contributing to the development or exacerbation of numerous health conditions, including cardiovascular disease, liver disease, cancer, diabetes, musculoskeletal disorders, infertility, mental health disorders, and sleep-related breathing disorders.

Also, in the text are not mentioned the effects of lipotoxicity in the context of depression, sleep apnea, infertility..which I would suggest to add and explain better with supported bibliography.

Response: Lipotoxicity, resulting from excessive lipid accumulation outside adipose tissues, extends beyond metabolic disorders like obesity and diabetes, impacting conditions such as depression, sleep apnea, and infertility [52]. In depression, lipotoxicity contributes to chronic inflammation and oxidative stress, disrupting neuronal function and neurotransmitter balance, potentially exacerbating depressive symptoms [53]. Sleep apnea, strongly linked to obesity, involves lipotoxicity-induced inflammation and oxidative stress, contributing to structural changes in airway tissues, leading to increased susceptibility to airway collapse during sleep [54]. In infertility, lipotoxicity disrupts endocrine function, altering sex hormone levels and insulin resistance, affecting ovarian and testicular function, impairing fertility. Lipid accumulation damages reproductive tissues, impacting gamete quality and function, hindering folliculogenesis, spermatogenesis, and embryo implantation [56]. 

Paragraph 2.3: I would add to this paragraph the link between efferocytosis and mitophagy in the context of immune response (macrophage and immune cells) to diseases. A more detailed explanation of the role of ROS in the context of lipotoxicity is missing. Also, refences are needed in lanes 208, 211, 215. I would also avoid to repeat the concepts already explained before like from lanes 221 to 226.

Response: Efferocytosis and mitophagy are two interconnected cellular processes involved in the regulation of immune responses, particularly in the context of macrophage and immune cell function during disease pathogenesis [56]. Efferocytosis and mitophagy play complementary roles in dampening inflammation and resolving immune responses [10]. Efferocytosis prevents the accumulation of apoptotic cells and the release of pro-inflammatory cellular contents, whereas mitophagy limits the release of DAMPs from dysfunctional mitochondria [10,57]. Together, efferocytosis and mitophagy contribute to the resolution of inflammation and the promotion of tissue repair following injury or infection [10].

Emerging evidence suggests that there is crosstalk between efferocytosis and mitophagy pathways, with shared molecular components and regulatory mechanisms [58]. For example, some molecules involved in the recognition and engulfment of apoptotic cells by macrophages, such as phosphatidylserine receptors and scavenger receptors, also play roles in the regulation of autophagy and mitophagy [7]. Furthermore, defects in efferocytosis or mitophagy can disrupt immune homeostasis and exacerbate inflammation, contributing to the pathogenesis of various diseases [10]. Dysregulated efferocytosis and mitophagy are implicated in the pathogenesis of numerous diseases, including autoimmune disorders, atherosclerosis, neurodegenerative diseases, and cancer [59]. In conditions where efferocytosis or mitophagy is impaired, such as in aging or chronic inflammatory diseases, the accumulation of apoptotic cells or dysfunctional mitochondria can promote inflammation and tissue damage [60].

detailed explanation of the role of ROS in the context of lipotoxicity

Response: Reactive oxygen species (ROS) are highly reactive molecules containing oxygen, such as superoxide anion (O2•−), hydrogen peroxide (H2O2), and hydroxyl radical (•OH). ROS are produced as natural byproducts of cellular metabolism, primarily in mitochondria, peroxisomes, and cytoplasmic enzymes like NADPH oxidases [61]. While ROS serve important roles in cell signaling, they can also lead to oxidative stress when produced in excess or when the cellular antioxidant defense mechanisms are overwhelmed [62]. In the domain of lipotoxicity, excessive lipid buildup in tissues beyond adipose can initiate mitochondrial dysfunction, inflammation, and oxidative stress, leading to an escalation in reactive oxygen species (ROS) generation [63].

Lipotoxicity often leads to mitochondrial dysfunction, characterized by impaired electron transport chain (ETC) activity, reduced ATP production, and increased electron leakage, resulting in elevated ROS generation within the mitochondria [63,64]. Excessive ROS production overwhelms the cellular antioxidant defense systems, including enzymes like superoxide dismutase (SOD), catalase, and glutathione peroxidase, as well as non-enzymatic antioxidants such as glutathione and vitamins C and E [65]. This imbalance between ROS production and antioxidant capacity leads to oxidative stress, causing damage to lipids, proteins, and DNA within the cell [66].

ROS can initiate lipid peroxidation, a chain reaction that damages cellular membranes by oxidizing polyunsaturated fatty acids (PUFAs) [67]. Lipid peroxidation generates lipid hydroperoxides and reactive aldehydes, such as malondialdehyde (MDA) and 4-hydroxynonenal (4-HNE), which can further exacerbate cellular damage and inflammation [39]. ROS serve as signaling molecules that activate inflammatory pathways, such as nuclear factor-kappa B (NF-κB) and mitogen-activated protein kinases (MAPKs), leading to the production of pro-inflammatory cytokines and chemokines [68]. Chronic inflammation further promotes lipotoxicity by perpetuating tissue damage and dysfunction [69].

Lipotoxicity-induced ROS production can also cause ER stress, leading to unfolded protein response (UPR) activation [70]. ER stress and UPR contribute to cellular dysfunction and apoptosis, further exacerbating tissue damage and dysfunction in lipotoxic conditions [70]. Prolonged exposure to elevated ROS levels can lead to cellular dysfunction and apoptosis, particularly in cells sensitive to oxidative stress, such as hepatocytes, cardiomyocytes, and pancreatic β-cells [71]. ROS play a central role in the pathogenesis of lipotoxicity by promoting mitochondrial dysfunction, oxidative stress, lipid peroxidation, inflammation, ER stress, cellular dysfunction, and apoptosis [72].

Top of Form

Paragraph 3.1: I would link the first 2 paragraphs as they repeat some concepts like apoptosis process. References needed for lanes 247, 258, 262, 265. Delete lane 278 (top of form)

Response: Section 3.1 rephrased. References added for lanes 247, 258, 262, 265. Lane 278 deleted

Paragraph 3.2: I would suggest to expand a bit this paragraph as secondary necrosis contributes to different diseases progression which are associated to impaired efferocytosis. References are needed in lanes 281, 283, 297, 304 and I suggest the author to avoid repeating the concept of apoptosis as already explained before (lane 288).

Response: Section 3.2 rephrased. References added for lanes 281, 283, 297, 304. Lane 288 deleted

Secondary necrosis, also known as post-apoptotic necrosis, refers to the process by which apoptotic cells undergo further degradation and disintegration following incomplete clearance by phagocytes [73]. When apoptotic cells are not efficiently engulfed and removed by phagocytic, they progress to secondary necrosis, leading to the release of cellular contents into the extracellular environment. This can trigger inflammation and contribute to the pathogenesis of various diseases, particularly those associated with impaired efferocytosis [74]. Examples of diseases where secondary necrosis contributes to disease progression due to impaired efferocytosis: In autoimmune diseases like systemic lupus erythematosus (SLE) and rheumatoid arthritis (RA), impaired efferocytosis results in the buildup of apoptotic cells, leading to secondary necrosis. This release of autoantigens activates autoreactive immune cells, worsening autoimmune responses and tissue damage [75]. In atherosclerotic plaques, deficient efferocytosis causes apoptotic cell accumulation, promoting secondary necrosis and releasing pro-inflammatory molecules, exacerbating plaque instability and atherosclerosis progression [76]. In chronic obstructive pulmonary disease (COPD), impaired efferocytosis leads to apoptotic cell buildup in the lungs, exacerbating inflammation, mucus production, and tissue remodeling. Secondary necrosis further worsens lung damage and inflammation [77]. In non-alcoholic fatty liver disease (NAFLD) and non-alcoholic steatohepatitis (NASH), impaired efferocytosis causes apoptotic hepatocyte accumulation, leading to secondary necrosis and releasing pro-inflammatory molecules, contributing to hepatic inflammation, fibrosis, and disease progression [78]. In neurodegenerative diseases such as Alzheimer's and Parkinson's, impaired efferocytosis results in apoptotic cell accumulation in the brain, triggering neuroinflammation and release of neurotoxic molecules, worsening neuronal damage and disease advancement [79]. Impaired efferocytosis can lead to the accumulation of apoptotic cells and secondary necrosis, contributing to inflammation, tissue damage, and disease progression in various pathological conditions, including autoimmune diseases, atherosclerosis, COPD, NASH, and neurodegenerative diseases [59].

Paragraph 3.3: References are missing in lanes 317, 319, 323, 324, 327, 330, 332

Response: References added

Delete lane 373

Response: Lane 373 deleted

 Paragraph 4: Add appropriate references in lanes: 376, 393, 398, 405, 412,430, 431, 432, 433, 453, 459, 464. Delete lane 473. Also, it would be nice to expand the concept of tumor immune-escape and how this could be associated to chronic inflammation and impaired efferocytosis and which strategies could be proposed for repristinate the immunological response of the body.

Response: Lane 473 deleted

Tumor immune escape refers to the ability of cancer cells to evade detection and elimination by the immune system, allowing them to proliferate and establish tumors. This process involves multiple mechanisms that suppress immune responses and promote tumor growth [80]. Chronic inflammation and impaired efferocytosis can contribute to tumor immune escape by creating an immunosuppressive microenvironment conducive to tumor progression [81].  Chronic inflammation is a hallmark of cancer development and progression. Inflammatory mediators produced in response to tissue damage, infection, or other insults can promote tumor initiation, angiogenesis, invasion, and metastasis [82]. Persistent inflammation in the tumor microenvironment leads to the recruitment of immune cells, such as tumor-associated macrophages (TAMs), myeloid-derived suppressor cells (MDSCs), and regulatory T cells (Tregs), which suppress anti-tumor immune responses and promote immune tolerance [43,83].

Impaired efferocytosis, the inefficient clearance of apoptotic cells, can contribute to chronic inflammation and immune dysregulation. Accumulation of apoptotic cells leads to secondary necrosis and the release of pro-inflammatory cytokines, chemokines, and damage-associated molecular patterns (DAMPs), which perpetuate inflammation and suppress anti-tumor immunity [10,84]. Additionally, impaired efferocytosis can lead to the accumulation of apoptotic tumor cells, which fail to induce effective anti-tumor immune responses and may even promote immunosuppression through the release of immunosuppressive factors [84]. Tumor cells employ various strategies to evade immune surveillance and elimination, including downregulation of major histocompatibility complex (MHC) molecules, expression of immune checkpoint molecules (e.g., PD-L1, CTLA-4), induction of immune tolerance, and recruitment of immunosuppressive cells [80]. These mechanisms inhibit effector T cell function, promote T cell exhaustion, and facilitate immune evasion, allowing tumor cells to evade immune destruction and proliferate unchecked [85].

To counteract tumor immune escape and restore anti-tumor immunity, several strategies can be proposed:

Enhancing Efferocytosis: Promoting efficient efferocytosis can help reduce inflammation and remove immunosuppressive apoptotic cells from the tumor microenvironment. Strategies to enhance efferocytosis include targeting efferocytosis receptors on phagocytes, modulating signaling pathways involved in efferocytosis, and promoting the resolution of inflammation [84].

Immune Checkpoint Inhibition: Blocking immune checkpoint molecules, such as PD-1/PD-L1 and CTLA-4, can restore anti-tumor immune responses and enhance T cell-mediated tumor killing. Immune checkpoint inhibitors (ICIs) have shown efficacy in a variety of cancers by unleashing the immune system to recognize and attack tumor cells [86].

Targeting Tumor-Associated Inflammation: Modulating the inflammatory microenvironment of tumors can enhance anti-tumor immunity and inhibit tumor growth. This can be achieved through targeting inflammatory mediators, such as cytokines and chemokines, or by repolarizing TAMs and MDSCs from immunosuppressive to anti-tumor phenotypes [87].

Immunotherapy: Various immunotherapeutic approaches, including cancer vaccines, adoptive cell therapy (e.g., CAR-T cells), and cytokine therapy, aim to boost anti-tumor immune responses and overcome immune evasion mechanisms employed by tumors. These strategies harness the power of the immune system to recognize and eliminate tumor cells [88].

Tumor immune escape is facilitated by chronic inflammation and impaired efferocytosis, which create an immunosuppressive microenvironment conducive to tumor growth [81]. Strategies aimed at restoring immunological responses, enhancing efferocytosis, and targeting tumor-associated inflammation can help counteract tumor immune escape and improve the efficacy of cancer immunotherapy [89].

Paragraph 5: References are missing throughout the paragraph, also I would suggest to add some examples of the different approaches that could be used for therapeutic intervention targeting efferocytosis and the possibility to perform personalized approaches.

Response: References added.

Table: Different approaches for therapeutic intervention targeting efferocytosis

Approach

Description

Personalized Aspect

Pharmacological Modulation

Development of pharmacological agents targeting efferocytosis receptors or signaling pathways involved in the recognition and engulfment of apoptotic cells.

Tailoring pharmacological interventions based on individual patient characteristics, such as genetic variations affecting efferocytosis receptors or underlying diseases.

Immunomodulatory Therapy

Utilization of immunomodulatory agents, such as cytokines, growth factors, or monoclonal antibodies, to enhance efferocytosis activity of immune cells.

Customizing immunomodulatory therapies based on the immune profile and responsiveness of individual patients, including considerations for immune cell function and cytokine levels.

Stem Cell Therapy

Administration of stem cells, such as mesenchymal stem cells (MSCs), which possess immunomodulatory properties and can promote efferocytosis by enhancing phagocytic activity of macrophages.

Selecting optimal stem cell sources and dosages based on individual patient characteristics, including age, underlying diseases, and immune status.

Gene Therapy

Genetic manipulation of efferocytosis-related genes or pathways using gene editing technologies, viral vectors, or RNA interference to enhance phagocytic capacity of immune cells.

Targeting specific genetic mutations or polymorphisms associated with impaired efferocytosis in individual patients and designing personalized gene therapy approaches.

Nutritional Interventions

Dietary interventions targeting nutrient deficiencies or imbalances that may impair efferocytosis function, such as omega-3 fatty acids, antioxidants, or vitamins.

Designing personalized nutrition plans based on individual dietary habits, nutritional status, and metabolic needs to optimize efferocytosis efficiency and overall immune function.

Lifestyle Modifications

Adoption of lifestyle habits, such as regular physical activity, stress management, and smoking cessation, which can positively influence immune function and efferocytosis activity.

Tailoring lifestyle recommendations to accommodate individual preferences, capabilities, and socioeconomic factors.

Combination Therapies

Integration of multiple therapeutic modalities to synergistically enhance efferocytosis and overall immune function.

Developing personalized combination therapy regimens based on individual patient profiles, including disease severity, treatment response, and potential drug interactions.

Top of Form

 Paragraph 6: This paragraph is very informative but lacks substantial references to support the authors claims.

Response: References added.

For Figure 3, I would suggest the authors to do a final figure where are listed all the different approaches to mitigate the risks associate to overnutrition and lipotoxicity

Response: Various approaches to mitigate the risks associated with overnutrition and lipotoxicity:

Approach

Description

Dietary Modifications

Emphasize a balanced diet rich in fruits, vegetables, whole grains, lean proteins, and healthy fats. Reduce intake of processed foods, sugary beverages, saturated fats, and trans fats. Monitor portion sizes and caloric intake.

Regular Physical Activity

Engage in regular aerobic exercise, strength training, and flexibility exercises. Aim for at least 150 minutes of moderate-intensity or 75 minutes of vigorous-intensity exercise per week. Incorporate physical activity into daily routines.

Weight Management

Maintain a healthy body weight through a combination of diet, exercise, and lifestyle modifications. Set realistic weight loss goals and seek support from healthcare professionals or support groups if needed.

Pharmacotherapy

Consider pharmacological interventions, such as anti-obesity medications or lipid-lowering drugs, under the guidance of a healthcare provider. May be recommended for individuals with obesity or dyslipidemia.

Bariatric Surgery

Surgical procedures such as gastric bypass, sleeve gastrectomy, or gastric banding may be considered for individuals with severe obesity or obesity-related comorbidities who have not responded to other weight loss interventions.

Nutritional Supplements

Consider supplementation with vitamins, minerals, and omega-3 fatty acids to address nutrient deficiencies and support overall health. Consult with healthcare provider or registered dietitian for personalized guidance.

Lifestyle Modifications

Adopt healthy lifestyle habits such as getting adequate sleep, managing stress, and avoiding smoking and excessive alcohol consumption. Practice mindful eating, focusing on hunger cues, portion control, and enjoyment of food.

Medical Monitoring

Regularly monitor blood glucose levels, lipid profiles, blood pressure, and other metabolic parameters to assess health status and track progress. Schedule routine check-ups with healthcare providers for personalized guidance.

Round 2

Reviewer 1 Report

Comments and Suggestions for Authors

Thank you for the revised manuscript with lots of effort. If you think that two high school students have contributed to your project and have spent enough time and effort for this manuscript, please give them credit as coauthors, which will be fair for them. I will support your decision, either way.